# An Overview of the Potential Use of Ethno-Medicinal Plants Targeting the Renin–Angiotensin System in the Treatment of Hypertension

**DOI:** 10.3390/molecules25092114

**Published:** 2020-04-30

**Authors:** Pietro De Lange-Jacobs, Asma Shaikh-Kader, Bianca Thomas, Trevor T. Nyakudya

**Affiliations:** 1Department of Human Anatomy and Physiology, University of Johannesburg, Doornfontein Campus, Corner Beit and Siemert Streets, Doornfontein, Johannesburg 2000, South Africa; pdljacobs@uj.ac.za (P.D.L.-J.); akader@uj.ac.za (A.S.-K.); bthomas@uj.ac.za (B.T.); 2Department of Physiology, School of Medicine, Faculty of Health Sciences, University of Pretoria, Pretoria 0002, South Africa

**Keywords:** hypertension, ethno-medicinal plants, renin–angiotensin system, ACE1, ACE2, Sub-Saharan Africa

## Abstract

The development of risk factors associated with cardiovascular disorders present a major public health challenge in both developed countries and countries with emerging economies. Hypertension and associated complications including stroke and myocardial infarction have reached pandemic levels. Current management strategies of hypertension predominantly include the utilization of pharmaceutical drugs which are often associated with undesirable side effects. Moreover, the drugs are often too expensive for populations from resource-limited Southern African rural, and some urban, communities. As a result, most patients rely on ethno-medicinal plants for the treatment of a variety of diseases including cardiovascular and metabolic disorders. The effectiveness of these plants in managing several cardiovascular diseases has been attributed to the presence of bioactive phytochemical constituents. In this review, the treatment options that target the renin–angiotensin system (RAS) in the management of hypertension were summarized, with special emphasis on ethno-medicinal plants and their influence on the ACE1 RAS pathway. The dearth of knowledge regarding the effect of ethno-medicinal plants on the ACE2 pathway was also highlighted.

## 1. Hypertension and the Global Problem

The worldwide incidence of cardiovascular and metabolic diseases is on the increase, as evidenced by the growing prevalence of hypertension, type 2 diabetes (T2DM), and obesity. In fact, the present epidemic of hypertension, obesity and metabolic disorders has been proposed as the primary cause for the lower life expectancy that has been predicted for the next generation [1]. A person is diagnosed as being hypertensive if they have a systolic blood pressure that is higher than 140 mmHg or a diastolic pressure that is above 90 mmHg [2]. Several factors such as the sex of an individual, age, obesity, urbanization and lifestyle choices (consumption of high-energy diet and physical inactivity) have consistently been associated with hypertension [3,4,5]. Hypertension remains a major problem in affluent Western societies, while its incidence is increasing in developing countries. The increasing prevalence of this non-communicable disease in developing countries has been shown to be associated with economic development, urbanization and lifestyle changes [4]. The prevalence of hypertension worldwide is predicted to increase from 26.4% in 2000 to 60% of the global population by 2025 [6]. Furthermore, a relatively recent epidemiology study conducted in 786 countries with 5.4 million participants suggested that the mortality rate attributed to hypertension was 7 million deaths per year worldwide [7]. 

Sub-Saharan Africa has a diversity of countries, ethnic groups and cultures with differences in socioeconomic status. According to [8], there may be significant variations in the prevalence of hypertension in urbanized, semi-urbanized, and non-industrialized, isolated rural areas. In lower income groups, the incidence of hypertension is believed to be increased by socio-economic stress and lack of exercise, while obesity, dietary excess, excessive alcohol consumption, and lack of exercise appear to increase the incidence of hypertension in higher income groups [8,9]. According to the African Regional Health Report [10], 80 million people suffered from hypertension in 2000 in Sub-Saharan Africa and the prediction is that by 2025, the figure will rise to 150 million. In South Africa, there are more than 6 million people with hypertension [11]. In fact, a recent study of 3820 participants older than 50 years found that approximately 78% suffered from hypertension, with a greater incidence in females (56%) as compared to males (44%) [4]. However, the risk of mortality from hypertension was greater in males than in females [10]. The rise in the incidence of hypertension in South Africa has been attributed to the rapid westernization characterized by the excessive consumption of high-energy diets and adoption of sedentary affluent lifestyles. The fact that one in four South Africans between the ages of 15 and 64 years suffer from hypertension reflects this and highlights the severity of hypertension in South Africa [5,12]. The prevalence of hypertension in the South African population may not necessarily be due to ethnic differences, but may be due to the differences in socio-demographic status [13]. However, Connor, et al. [14] reported that ethnicity affects the overall prevalence of hypertension in South Africa, with a higher incidence in the black population compared to other ethnic groups. In addition, black South Africans are twice as likely to suffer from strokes than white South Africans and appear to be more vulnerable to other cardiovascular complications such as heart failure [8,11]. It is possible that, in some ethnic groups, there is a decrease in awareness of the disease, leading to poor response and compliance with anti-hypertensive therapy. Poorly controlled hypertension contributes majorly to the general burden of diseases in the adult population and can lead to cerebrovascular diseases, myocardial infarctions, kidney disease or failure, as well as left ventricular hypertrophy, which could predispose to congestive heart failure. This can, however, be avoided by both prompt detection and cost-effective management of the condition [5]. Moreover, the epidemic of hypertension is confounded by the under-diagnosis of the disease [8]. It is therefore widely accepted that there is a dire need to improve the diagnosis and management of hypertension [8]. 

## 2. Pathways Involved in the Renin–Angiotensin System

The management of hypertension involves the use of pharmacological agents that target specific mechanisms, such as the renin angiotensin system (RAS), that are involved in the regulation of blood pressure. Examples of these include angiotensin-converting enzyme (ACE) inhibitors and beta-blockers. In the last twenty years, it was recognized that the classical RAS axis is much more complex than originally believed (Figure 1). This is due to the discovery of i) ACE2, which catalyses the formation of Angiotensin (Ang) (1-7) from Ang II [15,16] and ii) G-protein-coupled receptors for Ang (1-7) named Mas receptors [17] that have been found to mediate the vasodilatory effects of Ang (1-7). Although Ang (1-7) can also function via the angiotensin 2 (AT2) receptor to cause vasodilation, the affinity for these receptors appears to be very low [18]. Furthermore, Ang (1-7) antagonizes the effects of Ang II at the angiotensin 1 (AT1) receptors, while ACE inhibitors do not inhibit ACE2 [19]. ACE2 can also hydrolyze Ang I to Ang (1-9), which is consequently converted to Ang (1-7) by ACE. 

The ability of ACE2 to metabolize Ang II is crucial in the modulation or control of blood pressure and hypertension [20]. Due to the vasodilatory effects of Ang (1-7), it has been shown to be involved in the prevention of hypertension and cardiac hypertrophy [21] and the genetic deletion of Mas receptors have been shown to impair cardiac function [22]. Ang (1-7) is found in the myocardium, but is lost in infarcted hearts [23]. Studies have indeed shown that low doses of Ang (1-7) increase cardiac output and antagonize the vasoconstriction induced by Ang II by causing vasodilatory responses and reduced vascular resistance [24].

Since ACE2 is the main enzyme involved in counterbalancing the Ang II vasoconstrictory and Ang (1-7) vasodilatory effect [19], it is suggested to play a role in cardiovascular diseases (CVDs) such as hypertension. It is possible that ACE2 over-expression can shift the balance towards Ang-(1-7) production, consequently decreasing the Ang II available to cause hypertensive effects [20]. The beneficial effects of ACE2 may therefore not necessarily be its production of Ang (1-7) from Ang II, but its capacity to lower Ang II and thereby prevent the detrimental cardiovascular effects of Ang II [20].

Existing treatment options for hypertension include pharmaceutical drugs that target the RAS pathway such as angiotensin receptor antagonists, ACE inhibitors and ACE2 activators. However, current research also focuses on the potential therapeutic effect of medicinal plants on these pathways. 

## 3. Disadvantages of Pharmaceutical Drugs in the Management of Hypertension

Pharmaceutical management of hypertension is costly. In 1991, it contributed almost R5 billion in direct and indirect expenditure to the treatment of cardiovascular disease in South Africa [11]. Few people make the necessary lifestyle changes to improve their blood pressure, therefore most patients need antihypertensive medication for life. The consequence of this is two-fold: a decrease in the efficacy of antihypertensive medication due to prolonged usage and an increased incidence of adverse side effects such as coughing, loss of taste [25], erectile dysfunction [26], constipation and alopecia among others [27]. Although pharmaceutical treatment for hypertension exists, the inefficiency of the health care systems in Sub-Saharan Africa and the economic impact of expensive anti-hypertensive medication means that a large portion of the population find it challenging to adhere to or comply with the recommended treatment [5,28]. Furthermore, many hypertensive patients in developing countries of Sub-Saharan Africa do not have access to the expensive pharmaceutical anti-hypertensive medication due to several socio-economic challenges. It has also been shown that although ACE inhibitors have a good safety profile and are well tolerated, they are not safe to use during pregnancy [29].

As a result of these limitations associated with anti-hypertensive pharmaceutical products, and the fact that the incidence of hypertension is so rapidly increasing, there is an urgent need to find affordable, alternative or complementary treatments which are effective in reducing the development as well as the progression of hypertension and its symptoms. Of particular importance is prioritizing the research to develop an alternative or complementary approach that would increase the beneficial components of the RAS, especially the ACE2-Ang-(1-7)-Mas axis. Any safe, natural approach in the treatment of hypertension, which can be used with or without pharmaceutical drugs, is worth exploring. 

## 4. Use of Traditional Medicine in the Management of Hypertension

Medicinal plants possess health-promoting characteristics and over 80% of the population in the world’s developing countries rely on this medicinal source for their primary daily healthcare needs [30,31]. Evidence from literature shows that medicinal plants are used as a treatment option for a wide range of diseases, including hypertension and its complications [28]. The usage of medicinal plants for the treatment of hypertension has been documented in Asia, Europe, North America [32] and Africa [30].

The pharmacological and therapeutic properties of a number of medicinal plants that are currently used to manage hypertension by African communities are attributed to the presence of bio-active phytochemical constituents. These unique biologically active phytochemicals may improve health and alleviate the symptoms associated with hypertension either individually, additively or synergistically [30]. A review article by [28] on the use of medicinal plants in the treatment of hypertension in Sub-Saharan Africa and an ethno-pharmacological survey by Eddouks, et al. [33] have both highlighted the fact that scientifically sound information on the topic is scarce and further investigations in the bio-active compounds and action of the plant materials with anti-hypertensive properties need to be conducted. Although a few studies provide the very specific biological nomenclature for the plants used in the treatment of hypertension, (for example, in Nigeria, *Azadirachta indica*, *Aloe vera* and *Cassia alata* [28] and in Morocco, *Allium sativum*, *Olea europaea* and *Pimpinella anusum* [33]), the systematic identification of the medicinal plants used needs to be clarified. 

Sixty five percent of South African adults in rural areas still depend on traditional medicines for their primary healthcare, however the extent of its use is not well documented [5,28]. Since medicinal plants still form an integral part of the culture in rural South Africa, patients diagnosed with hypertension in medical clinics or hospitals often approach traditional healers for treatment [34]. The most popular species used as an indigenous treatment for hypertension include *Momordica balsamina*, *Hypoxis hemerocallidea* and *Aloe marlothii* [35]. In an experimental animal study, eight plant species in Kwa-Zulu Natal were identified to have anti-hypertensive properties, with *Tulbaghia violacea* (wild garlic) holding the most promise [36]. Somova et al. [37] also experimentally found that *Olea europaea subsp. africana* showed potential in hypertension treatment. Therefore, broadening our knowledge on the use of ethno-medicinal plants in the treatment of hypertension is of paramount importance.

## 5. Ethno-Medicinal Plants, Hypertension and the Effect on the ACE1 RAS Pathway

Known medicinal plants that are currently used, or have the potential to lower blood pressure by influencing the ACE1 RAS pathway, are summarized in Table 1. Most of the medicinal plants used to treat hypertension possess ACE inhibitor activity. Plants considered to possess potential antihypertensive properties are required to inhibit the ACE enzyme (and the subsequent conversion of angiotensin I to angiotensin II) by more than 50% [12,38]. Studies have shown that methanol and aqueous extracts of some traditional plants such as *Amaranthus hybridus*, *Amaranthus dubius*, *Justica flava*, *Asystasia gangetica*, and *Tulbaguia violacea*, among others, show significant ACE1 inhibitory activity [39]. 

Most research has been done on plants such as wild garlic, hibiscus, black cumin and chokeberry that are indigenous to the Old-World countries and, due to their known medicinal and culinary properties, have spread worldwide for centuries. In the last decade, interest in the use of medicinal plants indigenous to other continents has been on the rise and recent research indicates that several indigenous African plant species have the potential to lower hypertension by influencing the conventional RAS pathway. The plant species indigenous to Southern Africa that show the most promising effect on ACE1 inhibition include *Tulbaghia violacea* [12,36,40,41], *Aspalathus linearis* [42] and *Mesembryanthemum sp.* [12]. Flavonoids (sub-groups: flavones, flavanones, isoflavones, flavan-3-ols, anthocyanins and flavonols) are the largest group of polyphenolic compounds found in plants and were found to be the most common active compound in most of the plant species listed in Table 1. Other polyphenols such as tannins, allicin and terpenoids also have ACE1 inhibitory effects.

## 6. The ACE2 Pathway and the Effect on Hypertension

In vitro [58] and in vivo [59] studies have shown that stimulation of ACE2 expression in hypertensive animals prevents cardiac hypertrophy, suggesting that ACE2 has a potential cardioprotective function. ACE2 also lowers blood pressure by increasing the sensitivity of the baroreflex [20]. Ang II, on the other hand, has opposite effects to ACE2 and acts through AT1R to desensitize the baroreflex, increase arginine vasopressin (AVP) secretion and activate the sympathetic nervous system, causing an increase in blood pressure [60]. ACE2 is mainly expressed in the testis, heart and kidneys [15], as well as in the liver, intestines and lungs [61,62]. ACE2 expression has been observed in the brain [63] where it is presumed to function as a central regulator of heart function [64]. Over-expression of ACE2 in the brain has been shown to alleviate hypertension, while inhibition of ACE2 activity with MLN-4760, an ACE2 receptor blocker, has been shown to attenuate the baroreflex and increase blood pressure [64].

Since its discovery in 2000 [15,16] ACE2 has been targeted as a potential therapeutic agent for decreasing blood pressure and treating hypertensive patients. Evidence to support the potential beneficial role of ACE2 in CVDs comes from the fact that: (i) ACE2 activity and expression increase in the initial stages of heart failure, but decrease with the progression of the disease [65], (ii) over-expression of ACE2 in hypertensive rats infused with Ang II prevents cardiac hypertrophy [66], (iii) lower levels of ACE2 are shown to be associated with cardiac and renal pathologies in animal models [67] and (iv) over-expressed ACE2 gene transfer after neonatal development protects against elevated blood pressure and cardiac pathology in spontaneously hypertensive (SHR) models [59].

## 7. Therapeutic Substances Targeting the ACE2 Pathway

### 7.1. Pharmaceutical Drugs Prescribed for Hypertension Influencing the ACE2 Pathway

In the last decade, pharmacological research has identified active ingredients and developed drugs that influence the ACE2 pathway. These drugs and their mechanisms of action are summarized in Table 2.

### 7.2. Ethno-Medicinal Plants Influencing the ACE2 Pathway

*In vitro* and *in vivo* studies have been conducted to test the ACE inhibitory capacity of medicinal plants (Table 1), but there is a dearth of information on the potential effect of these plants on the ACE2-angiotensin-(1-7)-Mas receptor axis. Table 3 summarizes the ethno-medicinal plants that have been shown to work through this axis and, moreover, highlights the lack of knowledge and research on the effect of medicinal plants on this axis. Currently there is no evidence of research in this field in Sub-Saharan Africa.

## 8. Challenges and Future Prospects of Using Anti-Hypertensive Medicinal Plants

In light of the socio-economic challenges that societies in rural Southern Africa face, the use of ethno-medicinal plants is a cheaper and more accessible alternative to pharmaceutical anti-hypertensive therapeutic drugs [78]. There is also a general perception that medicinal plants are inherently safe and natural by those that utilize them for their primary healthcare and as anti-hypertensive remedies [79]. The use of medicinal plants is also associated with cultural preference and preservation of indigenous knowledge systems that have been passed down the generations [80,81].

Despite the numerous advantages associated with the use of anti-hypertensive medicinal plant products, there are several challenges that have been identified as well. Some of these challenges include the lack of standardization of plant preparations (for example, raw, boiled, and steeping), considering that different plant species have different potencies and will affect the dosage needed. Furthermore, the different parts of the plant used and the season during which the plant material is harvested must also be considered due to the differences in the phytochemical composition in different parts of these plants. In most cases, the use of plant products for the treatment of hypertension and other diseases, particularly in rural areas, is based on anecdotal evidence. Research targeted at the identification of active phytochemicals in medicinal plants seem to be the other viable option for supporting the efficacy claims for all anti-hypertensive medicinal plants. However, studies have shown that the type of extract used (aqueous or ethanolic) when determining the phytochemical composition of these plants yields different activity results [12]. Therefore, the methodology used in this research field needs to be standardized.

To our knowledge, with a few exceptions such as rooibos, most of the plant species currently used for medicinal purposes are not cultivated as crops but occur in the wild. However, it is possible for these plants to be contaminated with xenobiotic compounds found in pesticides through ground and surface water, soil and air [82]. In future, if undomesticated plants are commercialized for medicinal purposes, it is important that, for each species, the maximal residue levels (MRLs) for contaminants are determined [83].

The World Health Organization advocates for the use of medicinal plants and recommends further studies to identify the anti-hypertensive active phytochemical constituents of medicinal plants, quality standards and clinical efficacy [84]. Strong interdisciplinary collaboration that bridges the gap between modern biomedical medicine and traditional medicine is needed in order to develop effective medicine for the management of diabetes and cardiovascular diseases.

## 9. Conclusions

The epidemic of hypertension and the fact that there is an urgent need to improve the treatment of this disease is indisputable. Pharmaceutical drugs currently used in the treatment of hypertension often target the renin angiotensin system. Most inhibit the ACE1 pathway in some way; however, in recent years, a handful of drugs have been found to activate the ACE2 pathway, and in this way have a cardioprotective effect. Pharmaceutical drugs, however, pose certain challenges, such as the high cost and the resultant lack of availability to patients of a lower socioeconomic status, decreased efficacy with prolonged use, negative side effects and lack of safety of the use of certain drugs during pregnancy. Attention has therefore focused on the use of medicinal plants in the treatment of this disorder, with a number of plants showing anti-hypertensive properties through inhibition of the ACE1 pathway.

According to our knowledge, the effect of medicinal plants on the ACE2 pathway has not yet been elucidated. We believe that some of the phytochemicals found in traditional plants with anti-hypertensive properties may also be activating the ACE2 axis, leading to vasodilation and the subsequent decrease in total peripheral vascular resistance. However, it is clear from this review that scientifically sound information on the topic is scarce and further investigations into the active compounds and action of the plant materials with anti-hypertensive properties targeting the renin-angiotensin system are vital. We recommend further studies using appropriate animal models to assess the potential of ethno-medicinal plants in their use for the management of hypertension, particularly their potential in the ACE2-angiotensin-(1-7)-Mas receptor axis. The evaluation and authentication of traditional remedies, especially those used in the management of hypertension, will not only be invaluable for those who are dependent on traditional medicine to manage their hypertension, but this information could potentially prove to be useful for the future development of new pharmaceutical drugs. In fact, a significant proportion of pharmaceutical prescriptions currently in use are formulations derived from plants [85]. This may contribute to the formulation of an integrated healthcare system combining both ethno-medicinal and Western practices.

## Figures and Tables

**Figure 1 molecules-25-02114-f001:**
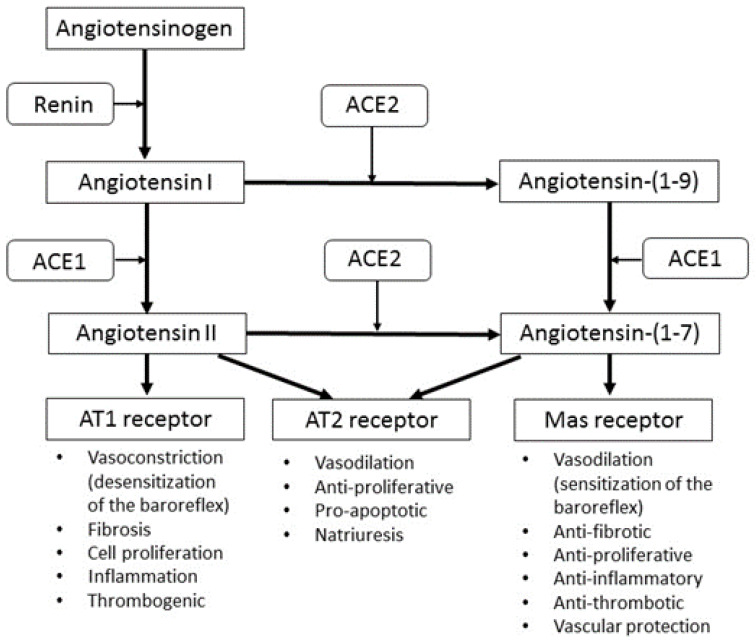
A current perspective of the ACE1 and ACE2 RAS pathways. ACE: angiotensin converting enzyme; AT: angiotensin.

**Table 1 molecules-25-02114-t001:** Ethno-medicinal plants used in the treatment of hypertension by influencing the ACE1 renin–angiotensin (RAS) pathway.

Species		Mechanism	Bioactive Phytochemicals	Geographical Distribution	Reference
* *Adenopodia spicata*	Spiny splinter bean	ACE1 inhibition	Flavonoids	Southern Africa	Duncan, Jäger and van Staden [12]
* *Agapanthus africanus*	African lily	ACE1 inhibition	Flavonoids Sitosterol, yuccagenin, agapanthagenin, spirostan sapogenins.	South Africa	Duncan, Jäger and van Staden [12]
* *Agave Americana*	Century plant, maguey, or American aloe	ACE1 inhibition	Flavonoids mono-2-ethylhexyl phthalate, 1,2-benzenedicarboxylic acid, n-docosane, and eicosane	Mexico, USA	Duncan, Jäger and van Staden [12]
*Allium sp.*	Wild Garlic	Decrease circulating angiotensin II	Allicin	Indigenous: Central Asia Currently found worldwide	Preuss, et al. [43]
* *Amaranthus dubius*	Red spinach, Chinese spinach, wild spinach	ACE1 inhibition	Flavonoids Niacin, thiamine, riboflavin, ascorbic acid, hydrocyanic acid, oxalic acid	Indigenous: China Currently found worldwide	Ramesar, Baijnath, Govender and Mackraj [38]
* *Amaranthus hybridus*	Smooth amaranth, smooth pigweed	ACE1 inhibition	Flavonoids, steroids, terpenoids, cardiac glycosides	North America	Ramesar, Baijnath, Govender and Mackraj [38]
*Aronia melanocarpa*	Chokeberry	Weak ACE1 inhibition	Polyphenols	North America	Sikora, et al. [44]
*Aspalathus linearis*	Rooibos	ACE1 inhibition	Flavonoids, polyphenols	Western Cape, South Africa	Persson [42]
* *Asystasia gangetica*	Creeping foxglove	ACE1 inhibition	Flavonoids Alkaloids, terpenes, salidroside, apigenin, ajugol, megastigmaneglucoside, benzyl β-D-glucopyranoside	Tropics	Ramesar, Baijnath, Govender and Mackraj [38]
*Berberis integerrima*	Barberry	ACE1 inhibition	Flavonoids, flavinols, flavonols, anthocynins, isoflavones, flavones, and other phenolic compounds.	Iran	Kearney, Whelton, Reynolds, Muntner, Whelton and He [6]
*Caragana microphylla*	Littleleaf Peashrub	ACE1 inhibition	Flavonoids, flavinols, flavonols, anthocynins, isoflavones, flavones, and other phenolic compounds.	Mongolia, China	Kearney, Whelton, Reynolds, Muntner, Whelton and He [6]
*Cecropia glaziovii*	Pumpwood (*guarumo*)	ACE1 inhibition	Flavonoids and proanthocyanidins	S. America-southern and eastern Brazil	Lacaille-Dubois, et al. [45]
*Crataegus spp*	Hawthorn	Weak ACE1 inhibitory effect	Bioflavonoids and proanthocyanidins	North America and Europe	Brixius, et al. [46] Rawat, et al. [47]
* *Dietes iridioides*	African iris, Cape iris	ACE1 inhibition	Flavonoids	Sub-Saharan Africa	Duncan, Jäger and van Staden [12]
*Galinsoga parviflora*	Potato weed	ACE1 inhibition	Not known	Central America. Currently found worldwide	Ramesar, Baijnath, Govender and Mackraj [38]
*Guazuma ulmifolia*	West Indian elm	Inhibits binding of Ang II to angiotensin II type 1 receptor	Proanthocyanidins	Central America	Caballero-George, et al. [48]
*Hibiscus sabderiffa*	Hibiscus	ACE1 inhibition	Anthocyanins (delphinidin-3-sambubiocyde and cynadine-3-sambubiocyde)	Africa, South East Asia, and Central America	Ojeda, et al. [49]
*Ipomoea reniformis*	Morning glory	ACE1 inhibition	Polyphenols	Subcontinent of Asia, China, Indonesia, Australia and Africa	Jabeen and Aslam [50]
* *Justicia flava*	Water-willow and shrimp plant	ACE1 inhibition	Flavonoids Sterols, salicyclic acid, lignins, docosanoic acid	tropical to warm temperate regions worldwide	Ramesar, Baijnath, Govender and Mackraj [38]
* *Mesembryanthemum sp.*	Fig marigold or Icicle plant	ACE1 inhibition	Flavonoids Betanidin, isobetanin, sterols, sapogenines, triterpenes, tannins and alkaloids	Southern Africa	Duncan, Jäger and van Staden [12]
*Musanga cecropioides*	Corkwood	ACE1 inhibition	Flavonoids and proanthocyanidins	Tropical Africa	Lacaille-Dubois, Franck and Wagner [45]
*Nigella sativa*	Black cumin	ACE1 inhibition	Thymoquinone and polyphenols	Middle East, India and Northern Africa	Jaarin, et al. [51]
*Nymphaea alba*	White waterlily	ACE1 inhibition	Flavonoids, flavinols, flavonols, anthocynins, isoflavones, flavones, and other phenolic compounds.	Europe, Middle East and North Africa	Kearney, Whelton, Reynolds, Muntner, Whelton and He [6]
*Ocimum gratissimum*	African basil	ACE1 inhibition	Phenolic compound, rutin	Africa, Madagascar, Southern Asia Naturalized in the West Indies and surrounding countries	Shaw, et al. [52]
*Olea europaea subsp. Africana*	Wild olive	ACE1 inhibition	Oleuropein, esculin, ursolic acid, scopolin and oleanolic acid	Africa	Msomi and Simelane [53]
*Onopordum acanthium*	Cotton thistle	ACE1 inhibition	Flavonoids, flavinols, flavonols, anthocynins, isoflavones, flavones, and other phenolic compounds.	Europe, northern Africa, the Canary Islands, the Caucasus, and southwest and central Asia.	Kearney, Whelton, Reynolds, Muntner, Whelton and He [6]
*Oxygonum sinuatum*	Double thorn	ACE1 inhibition	Not known	Sub-Saharan Africa	Ramesar, Baijnath, Govender and Mackraj [38]
*Peperomia pellucida*	Shiny bush	ACE1 inhibition	Terpenoids, Glycosides, Antraquinones, Tannins	Indonesia	Saputri, et al. [54]
*Physalis viscosa*	Sticky gooseberry	ACE1 inhibition	Not known	South America, Naturalised world-wide	Ramesar, Baijnath, Govender and Mackraj [38]
* *Protorhus longifolia*	Red beech	ACE1 inhibition	Flavonoids Triterpenes	South Africa, Swaziland	Duncan, Jäger and van Staden [12]
*Quercus infectoria*	Aleppo oak	ACE1 inhibition	Flavonoids, flavinols, flavonols, anthocynins, isoflavones, flavones, and other phenolic compounds.	Greece, Asia minor	Kearney, Whelton, Reynolds, Muntner, Whelton and He [6]
*Rosa rugose*	Beach rose	ACE1 inhibition	Not known	East Asia Naturalised in Europe and North America	Xie and Zhang [55]
*Rubus sp.*	Berries	ACE1 inhibition	Flavonoids, flavinols, flavonols, anthocynins, isoflavones, flavones, and other phenolic compounds.	North America, Europe	Kearney, Whelton, Reynolds, Muntner, Whelton and He [6]
*Tulbaghia violacea*	Garlic	Reduced BP–ACE1 and β1 inhibition (may not act via AT1 receptors or α1 receptors)	Bioflavonoids, steroidal saponins	South Africa, Zimbabwe	Duncan, Jäger and van Staden [12] Mackraj, Ramesar and Singh [36] Raji, Mugabo and Obikeze [40] Raji, Obikeze and Mugabo [41] Ramesar, Baijnath, Govender and Mackraj [38]
*Viola mandshurica*	Manchurian violet	ACE1 inhibition	Not known	East Asian region, China, Taiwan, Mongolia, Japan, Russia and the Far East	Huh [56]
Zingiber ottensii	Red beehive ginger	ACE 1 inhibition	Bioactive protein peptides	Tropics of Africa, Asia and the Americas	Yodjun, et al. [57]

* Plants aqueous or ethanolic extracts inhibited the ACE enzyme by greater than 50% and tested negative for tannins using the gelatin salt block test.

**Table 2 molecules-25-02114-t002:** Pharmaceutical drugs targeting the ACE2 pathway.

Category of Action	Mechanism of Action	Examples of Drugs/Active Ingredient	References
1. ACE2 activators	Decrease expression of ACE	Felodipine combined with puerarin *	Bai, et al. [68]
		Ibuprofen #	Qiao, et al. [69]
	Decrease expression of AT1R	Felodipine combined with puerarin *	Bai, Huang, Chen, Wang and Ding [68]
		Ibuprofen #	Qiao, Wang, Chen, Zhang, Liu, Lu, Guo, Yan, Sun and Hu [69]
	Decrease expression of serum Ang II	Felodipine combined with puerarin *	Bai, Huang, Chen, Wang and Ding [68]
		Ibuprofen #	Qiao, Wang, Chen, Zhang, Liu, Lu, Guo, Yan, Sun and Hu [69]
	Increase expression of ACE 2	Candesartan CILEXETIL #	Arumugam, et al. [70]
		Felodipine combined with puerarin *	Bai, Huang, Chen, Wang and Ding [68]
		Ibuprofen #	Qiao, Wang, Chen, Zhang, Liu, Lu, Guo, Yan, Sun and Hu [69]
		Propofol *	Cao, et al. [71]
		Telmisartan #	Sukumaran, et al. [72]
		Xanthenone and resorcinolnaphthalein *	Hernández Prada, et al. [73]
	Increase expression of Ang-(1-7)	Candesartan CILEXETIL #	Arumugam, Thandavarayan, Palaniyandi, Giridharan, Arozal, Sari, Soetikno, Harima, Suzuki and Kodama [70]
		Felodipine combined with puerarin *	Bai, Huang, Chen, Wang and Ding [68]
		Ibuprofen #	Qiao, Wang, Chen, Zhang, Liu, Lu, Guo, Yan, Sun and Hu [69]
		TBTIF (4-tert-butyl-2,6-bis(thiomorpholin-4-ylmethyl)phenol) #	Flores-Monroy, et al. [74]
	Increase expression of Mas receptor	Candesartan CILEXETIL #	Arumugam, Thandavarayan, Palaniyandi, Giridharan, Arozal, Sari, Soetikno, Harima, Suzuki and Kodama [70]
		Felodipine combined with puerarin *	Bai, Huang, Chen, Wang and Ding [68]
		Ibuprofen #	Qiao, Wang, Chen, Zhang, Liu, Lu, Guo, Yan, Sun and Hu [69]
		Telmisartan #	Sukumaran, Veeraveedu, Gurusamy, Lakshmanan, Yamaguchi, Ma, Suzuki, Kodama and Watanabe [72]
2. Ang-(1-7) Mas receptor agonist	Increase activation of Mas receptor	AVE 0991 #	Santos, Ferreira and e Silva [22]

* Hypertension; # Cardiovascular disease.

**Table 3 molecules-25-02114-t003:** Ethno-medicinal plants influencing the ACE2 pathway.

Species	Mechanism	Bioactive Phytochemicals	Geographical Distribution	Reference
*Qishenyiqi* (*Radix Astragali Mongolici*, *salvia miltiorrhiza bunge*, *Flos Lonicerae*, *scrophularia*, *Radix Aconiti Lateralis Preparata*, *Radix Glycyrrhizae*)	Blocks effect of Ang II by acting on AT1R and AT2 RIncreases ACE2	Not known	East Asia	Wang, et al. [75]
*Rosmarinus officinalis* Linn.	Decreases ACE expression and increases ACE2 expression, decreases expression of AT1R	Rosmarinic acid	Mediterranean region, but widely cultivated	Liu, et al. [76]
*Scutellaria baicalensis*	Enhances ACE2 protein expression	Flavonoids (Baicalin)	North America	Zhang, et al. [77]

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
