# Peer review of "An Overview of the Potential Use of Ethno-Medicinal Plants Targeting the Renin–Angiotensin System in the Treatment of Hypertension"

_molecules, 2020, doi:10.3390/molecules25092114_

Round 1

Reviewer 1 Report

The manuscript summarized that target the renin-angiotensin system (RAS) in the management of hypertension, with special emphasis on ethno-medicinal plants and their influence on the ACE1 RAS pathway. There are some questions need to be responded before the manuscript been considered for publication.

Q1: Despite there are many advantages of using ethno-medicinal plants targeting the renin-angiotensin system in the treatment of hypertension, try to discuss the standardization of pesticide residues in ethno-medicinal plants. Please refer to WHO. 2007, Quinn et al ., 2011; Kumar et al. 2018.

Q2:  Lane 167, alcohol should be corrected to methanol.

Author Response

General comment

The manuscript summarized that target the renin-angiotensin system (RAS) in the management of hypertension, with special emphasis on ethno-medicinal plants and their influence on the ACE1 RAS pathway. There are some questions need to be responded before the manuscript been considered for publication.

Comment 1

Q1: Despite there are many advantages of using ethno-medicinal plants targeting the renin-angiotensin system in the treatment of hypertension, try to discuss the standardization of pesticide residues in ethno-medicinal plants. Please refer to WHO. 2007, Quinn et al ., 2011; Kumar et al. 2018.

Response 1

We have added in the following paragraph to section 8 in response to the reviewer’s comment.

“To our knowledge, with a few exceptions such as rooibos, most of the plant species currently used for medicinal purposes are not cultivated as crops but occur in the wild. However, it is possible for these plants to be contaminated with xenobiotic compounds found in pesticides through ground and surface water, soil and air (Kumar et al., 2018). In future, if undomesticated plants will be commercialized for medicinal purposes, it is important that for each species, the maximal residue levels (MRLs) for contaminants, be determined (Quinn et al., 2011)” See Lines 244-249. 

Comment 2

Q2:  Lane 167, alcohol should be corrected to methanol.

Response 2

Alcohol has been corrected to methanol as suggested by the reviewer. See line 168.

Reviewer 2 Report

This is a nice comprehensive review article on the use of medicinal plants acting in RAS to treat hypertension. Overall the manuscript insofern high quality and high interest to the field.

there are just some minor aspects I could find which might be improved:

  1. figure 1. List of effects on the bottom left. For unification purposes, use maybe „thrombosis“ , „Platelet aggregation“or similar term instead of „thrombogenic“

2. line 142 , Space is missing between reference 28 and the following text

3. table 1. Hibiscus sabderiffa. „ACE inhibition“ is shifted one line up and does not match the line

Author Response

Comments and Suggestions for Authors

General Comment

This is a nice comprehensive review article on the use of medicinal plants acting in RAS to treat hypertension. Overall the manuscript is of a high quality and high interest to the field. There are just some minor aspects I could find which might be improved:

Comment 1

Figure 1. List of effects on the bottom left. For unification purposes, use maybe “thrombosis”, “Platelet aggregation: or similar term instead of  “thrombogenic”

Response 1

We have considered the reviewer’s suggestion and we do not feel that it is necessary to change thrombogenic to the suggested phrases.

Comment 2

Line 142 , Space is missing between reference 28 and the following text

Response 2

The space has been added as suggested by the reviewer. See line 142.

Comment 3

Table 1. Hibiscus sabderiffa. “ACE inhibition” is shifted one line up and does not match the line

 Response 3

Hibiscus sabderiffa has been adjusted in the table as suggested by the reviewer. See Table 1.

Round 2

Reviewer 1 Report

In the revised manuscript, My questions in experimental design has been responded by authors. Thus I suggest that the manuscript be considered for publication.